# Relative Effectiveness of Amorphous Silica, Malathion, and Pirimiphos Methyl in Controlling *Sitophilus oryzae* and *Tribolium castaneum* and Their Long-Term Effects on Stored Wheat Under Laboratory Conditions

**DOI:** 10.3390/insects16090981

**Published:** 2025-09-19

**Authors:** Nawal Abdulaziz Alfuhaid, Mohamed S. Shawir

**Affiliations:** 1Department of Biology, College of Science and Humanities in Al-Kharj, Prince Sattam Bin Abdulaziz University, Al-Kharj 11942, Saudi Arabia; 2Department of Pesticide Chemistry and Technology, Faculty of Agriculture, Alexandria University, El-Shatby, Alexandria 21545, Egypt; shawir56@gmail.com

**Keywords:** amorphous silica, *S. oryzae*, *T. castaneum*, residual toxicity, stored wheat

## Abstract

Pests such as weevils and beetles often attack stored grains like wheat, leading to significant food losses. This study tested different ways to control these pests using natural powders made from silica (a sand-like material) and two typical chemical insecticides. The aim was to find safer and longer-lasting solutions to protect stored wheat. The researchers found that one type of silica powder, with very tiny particles, was especially effective in killing pests, especially in dry conditions. However, the powders did not work well when the grain was moister. Interestingly, when the chemical insecticides were mixed with silica powder, their pest-killing power increased and lasted longer, even during long-term storage. One mixture, pirimiphos methyl combined with silica, remained very effective for over six months. These findings suggest that using natural silica dust, especially in dry storage conditions, can improve pest control while reducing the need for strong chemicals. These findings may help farmers and storage operators reduce postharvest losses and support safer, more environmentally friendly pest management practices.

## 1. Introduction

Wheat (*Triticum aestivum* L.) provides about 20% of global daily calories and protein and is a staple for over one-third of the world’s population [1,2]. In Egypt, annual production reaches 9.8 million metric tons, yet heavy reliance on imports highlights the need to protect local reserves [3]. Post-harvest losses from insect infestations are significant, especially under the country’s warm, humid storage conditions. While global losses of stored grain reach 20–30%, Egypt experiences 10–15% reductions in quantity and quality [4].

The most destructive pests are the rice weevil, *Sitophilus oryzae* (L.), and the red flour beetle, *Tribolium castaneum* (Herbst). The weevil bores into whole kernels, whereas the beetle infests processed grain, releasing toxic quinones that reduce nutritional value, grain weight, and marketability while fostering mold and mycotoxin contamination [5].

Chemical insecticides like malathion and pirimiphos methyl have historically been employed to protect stored products. Malathion has high toxicity and is economically inexpensive for controlling storage insect pests; however, it has low residues, and insects may develop resistance to it [6,7]. Pirimiphos methyl, however, exhibits extended residual efficacy, lasting for as long as 12 months under optimal conditions [8]. However, increasing concerns about pesticide residues, resistance, and environmental and human health risks have necessitated the search for better alternatives [9]. Amorphous silica, a desiccant powder with low toxicity to mammals, has emerged as a promising physical control agent. It exterminates insects by compromising their cuticle, resulting in fatal desiccation, and presents no threat of resistance emergence [10,11]. Furthermore, its efficacy can be enhanced with small amounts of synthetic insecticides, making it appropriate for integrated pest management (IPM) programs [12]. This study aims to assess the comparative efficacy and synergistic effects of amorphous silicas, malathion, and pirimiphos methyl against *S. oryzae* and *T. castaneum*, and evaluate their residual effectiveness during extended storage under laboratory conditions to underscore the potential of eco-friendly alternatives such as amorphous silica in promoting sustainable, low-risk pest management strategies and mitigating wheat losses.

## 2. Materials and Methods

### 2.1. Materials

#### 2.1.1. Insects and Cultures

The stored-product insects employed in this study were the rice weevil, *S. oryzae* (L.) (Coleoptera: Curculionidae), and the red flour beetle, *T. castaneum* (Herbst) (Coleoptera: Tenebrionidae), from stock cultures that had been maintained in our laboratory for many years without exposure to insecticides. The rice weevil, *S. oryzae*, was reared on sterilized whole wheat. Whole wheat samples were placed in glass jars, covered, and sterilized by heating at 70 °C for 1 h. The wheat was cooled and allowed to reabsorb water before use. It was then transferred to separately sterilized glass culture jars (capacity: 1 L; dimensions: approximately 10 cm diameter × 15 cm height) to a depth of 5 cm using approximately 250 g of wheat per jar, and a band of fluon was painted around the inside of the jar to prevent the insects from escaping. A total if 200–400 adult weevils from a previous culture were added, and the mouth of the jar was sealed with filter paper stuck down with glue. After two weeks, the insects were sieved out of the mixture, discarded or transferred to another jar, and the jar resealed. The insects were reared at 25 ± 1 °C and 70 ± 5% R.H.; under these conditions, the period from egg lay to adult emergence was about 5 weeks. Adult insects (2–3 weeks after emergence) were used for bioassay experimental work. The red flour beetle, *T. castaneum*, was reared on a mixture of locally purchased whole meal flour, bran, and yeast (18:17:1 *w*/*w*/*w*). Flour and bran were sterilized by heating at 70 °C for one hour, allowed to cool and reabsorb water, then mixed with the dried yeast. The mixture was added to sterilized glass culture jars to a depth of 5 cm. Two hundred thousand four hundred adults from a previous culture were added, and the mouth of the jar was sealed with filter paper stuck down with glue. After 2 weeks, the insects were sieved out of the medium, discarded, or transferred to another jar, and the jar was resealed. The insects were cultured in complete darkness under the same temperature and relative humidity conditions. Under these conditions, the period from egg lay to emergence was approximately 7 weeks. The mixed gender insects were used for bioassays 2–3 weeks following emergence.

#### 2.1.2. Chemicals

Nine amorphous dusts were used in this study, including two fumed silicas (high-purity silica, ~99.8% SiO_2_, and hydrophilic fumed silica), six precipitated silicas (all amorphous types with different physical characteristics and surface areas), and one mineral dust (talc, magnesium silicate hydroxide, ~98% purity).

#### 2.1.3. Insecticides

Technical-grade malathion (85.4% *w*/*w*) and pirimiphos methyl (94.5% *w*/*w*) were procured from certified suppliers.

### 2.2. Methods

#### 2.2.1. Insecticides Formulated on Dusts

A 0.2% *w*/*w* formulation of malathion on silica was prepared by dissolving 9.796 mg of technical grade malathion in hexane, adding the silica dust (4.169 g), stirring for one hour with a magnetic stirrer, and then evaporating to dryness using the rotary evaporator. Formulations of pirimiphos methyl (0.1% *w*/*w*) on silica and formulations of malathion 4% and pirimiphos 2% on talc dust were prepared similarly.

#### 2.2.2. Admixing Dusts with Grain

The untreated wheat was dried in an incubator at 35 °C and conditioned to a 12% and 15% moisture content. A calculated amount of water was mixed with 4 kg batches. The amount of water (G mL) required to bring 100 g of grain to a moisture content of M2% (wet weight) from an initial moisture content of M1% was calculated using the formula of Oxley and Pixton [13].$$G=\frac{100\left(M2-M1\right)}{\left(100-M2\right)}$$

The batches were stored in airtight containers at 5 °C for at least 2 weeks before use. Wheat moisture contents were determined by using the AOAC air oven method [14], with the mean of three sample determinations being taken. Wheat (100 g) batches at 12% and 15 ± 0.1% moisture content were added to 1 lb jam jars. Dusts were separately weighed onto aluminum foil, which was inverted over the mouth of the jar and tapped to remove dust. The jars were sealed with an adhesive plastic film and lid. Dusts were mixed with grain by holding the jar horizontally and rotating it around its vertical axis for one minute, while shaking it vigorously at the following intervals: 0–5 s, 20–25 s, and 40–45 s. After mixing, the jars were kept sealed until the dust had settled on the wheat. A band of fluon was then painted around the inside of the jar above the wheat to prevent the weevils from escaping. Fifty mixed-sex adult insects were introduced per replicate jar. Mortality was recorded after 3 days of incubation at 25 °C.

#### 2.2.3. Method of Bioassays

After three days of exposure, the treated wheat and beetles were tipped onto a tray, and the beetles were separated. Dead beetles were those that appeared brittle and did not move during a 2 min observation period. The mortality of each concentration was calculated after three days as the mean of three replicates. LC_50_ values of the probit mortality/log (dose) regression and associated statistics were computed using Finney’s method [15].

#### 2.2.4. Adsorption Isotherm of Silica Used for Bioassays

Samples of the different silicas (1 g) were weighed into aluminum foil dishes (2″ diameter × 1.5″ depth). The dusts were then exposed to relative humidities ranging between 40 and 100% for two weeks by incubating the dust over saturated salt solutions at 25 °C [16] as follows: K_2_CO_3_ (42.8% r.h.), Mg(NO_3_)_2_ (52.9% r.h.), NaBr (57.5% r.h.), SrCl_2_ (70.8% r.h.), NaCl 1% r.h.), KBr (80.7% r.h.), and BaCl_2_ (90.2% r.h.). At the end of each exposure period, the dusts were weighed, then dried at 110 °C for an hour and reweighed to determine the amount of water adsorbed.

#### 2.2.5. Residual Efficacy of Pirimiphos Methyl, Malathion, and Amorphous Silica Against *T. castaneum* and *S. oryzae*

Formulations of malathion and pirimiphos methyl on silica (Sipernat 22) were prepared using the method previously described, using technical-grade active ingredients. The wheat was “organic” and was conditioned to 12.0% and 15.0% moisture content before use. The rates of treatment of the wheat with dust formulations were as follows: pirimiphos methyl (2% dust), 4 mg a.i./kg; malathion (4% dust), 8 mg a.i./kg; silica, 0.47 g/kg (the estimated LC_50_ for *S. oryzae* at 12% m.c. and 25 °C); malathion/silica formulation, 0.47 g/kg (corresponding to 9.3 mg malathion a.i./kg); pirimiphos methyl/silica formulation, 0.47 g/kg (corresponding to 4.4 mg pirimiphos methyl a.i./kg). The wheat was treated in 4 kg batches, and 100 g samples were stored in sealed jars at 25 °C for varying periods up to 25 weeks after treatment. *T. castaneum* and *S. oryzae* were cultured as previously described and stored as mixed-sex populations from 2–3 weeks after adult emergence. Fifty insects were added to jars and incubated at 25 °C for 72 h before mortality determination for each storage period. The bioassay was carried out in triplicate.

### 2.3. Statistical Analysis

The mortality percentages were subjected to probit analysis according to Finney 15 to obtain the LC_50_ values and confidence limits using SPSS 12.0 (SPSS, Chicago, IL, USA). The LC_50_ values were considered significantly different if the 95% confidence limits did not overlap. In addition, quantitative data were analyzed using one-way ANOVA, and means were separated using the LSD test at a 5% significance level. Means within columns sharing the same letter were not significantly different [17].

## 3. Results and Discussion

### 3.1. Physical Properties of Different Dusts and Their Effects on S. oryzae

Fumed silicas (Cab-O-Sil EH5, Wacker HDK H20) feature extremely low bulk densities, very fine particle sizes, high surface areas, and strong oil adsorption capacities that enhance their ability to desiccate insects. Precipitated silicas exhibit greater variability, typically possessing higher bulk densities and larger particle sizes. Certain types (e.g., G23D) retain elevated surface areas, compromising effectiveness and manageability. Sipernat 22 has intermediate characteristics with notable adsorption capabilities. Conversely, talc possesses coarse particles and exhibits low adsorption, restricting its insecticidal efficacy. The comparative physical properties influencing these effects are summarized in Table 1.

### 3.2. Effect of Different Dusts on S. oryzae at 12% Grain Moisture Content and 25 °C

The tested amorphous silicas exhibited varying degrees of insecticidal activity against *S. oryzae* under laboratory conditions (12% grain moisture content and 25 °C, with 3-day exposure). The most toxic fumed silica was Wacker HDK H20 (LC_50_ = 19.4 mg/100 g), followed by Cab-O-Sil EH5 (LC_50_ = 41.0 mg/100 g). These materials possess extremely fine particle sizes (~0.005–0.007 µm), enhancing their desiccating ability, although they may pose operational challenges, such as poor flowability and airborne dust hazards. Among the precipitated silicas, GBBC and G937 demonstrated moderate efficacy, with LC_50_ values of 31.3 and 34.6 mg/100 g, respectively. Sipernat 22 showed the lowest insecticidal activity (LC_50_ = 46.6 mg/100 g) but presents practical advantages as a carrier due to its consistent performance and lower handling risks. The data emphasize that particle size and silica structure critically affect insecticidal efficiency. The regression analysis results and LC_50_ values are summarized in Table 2.

### 3.3. Effect of Different Dusts on S. oryzae at 15% Grain Moisture Content and 25 °C

The efficacy of Wacker HDK H20 in fuming silica was noticeably higher (LC_50_ = 47.1 mg/100 g). On the other hand, several others such as GBBC, GBBN, and Sipernat S22 showed higher LC_50_ values (between 70 and 83 mg/100 g), indicating decreasing potency. Sipernat S22 was more suitable as a carrier than a primary insecticidal dust because of its consistent efficacy across doses, as demonstrated by its reliable regression fit.

Table 3 shows the efficacy of all assessed amorphous silica dusts against *S. oryzae* at 15% grain moisture content and 25 °C. Compared to lower moisture levels (12%), the LC_50_ performance values were higher. The decrease in activity is probably due to the low desiccation effects in humid environments, which may lessen dust adhesion to insect cuticles. The current findings demonstrated that grain moisture content affected the effectiveness of silica dusts. Fumed silicas at 12% and/or 15% grain moisture contents were significantly more effective compared with precipitated silicas, which might be due to their very fine particles, high surface area, and strong oil adsorption capacity, which enhance insect mortality through cuticle abrasion and dehydration [18]. The results match those stated by Wang et al. [19], indicating the improved effectiveness of nano-sized silica (≤30 nm) in arid conditions. Zing Zing et al. [20] found that the bio-silica derived from rice husk showed consistent insecticidal efficacy in humid conditions.

Additionally, according to Superfine [21], the effectiveness of inert dusts is influenced by grain moisture content; 12–14% moisture is the ideal range for control. Consequently, particle size and grain moisture are essential factors that should inform the selection and formulation of silica-based protectants for managing pests in stored grain. Furthermore, many researchers have examined the effectiveness of inert dusts against *S. oryzae* in stored grains. Diatomaceous earth (DE) and kaolin have shown significant efficacy, resulting in up to 98% death of *S. oryzae* adults within 2–7 days at elevated concentrations [22]. DE consistently exceeded other inert dusts and botanical powders in managing *S. oryzae* [23].

### 3.4. Water Adsorption Isotherm of Various Amorphous Silica Used for Bioassays

Figure 1 illustrates how water adsorption increases with relative humidity (RH). This trend is consistent with the physical adsorption principle, where higher humidity provides more water molecules for surface adsorption. Among the amorphous silicas, G35 showed the highest water adsorption at 90.2% relative humidity (40.7%), followed by GBBC (28.8%) and G23D (29.3%), expected due to their enhanced surface area, pore volume, and abundance of surface silanol groups that increase hydrophilicity [24]. These silicas may exhibit significant effectiveness in arid conditions but may lose efficacy in humid environments due to particle aggregation and reduced abrasiveness [25]. Moderate adsorption was observed in Cab-BH5, Sipernat S22, and GBBN; for instance, S22 demonstrated a controlled increase in water absorption (4.40% to 10.05% across varying relative humidity levels), reflecting a balanced performance between activity and moisture resistance [26]. Conversely, Wecker H_2_O and Talc demonstrated minimal adsorption, with Talc attaining a high of only 0.20% at 90.2% relative humidity, indicating its hydrophobic properties. These silicas have enhanced stability in humid environments and may retain insecticidal properties more efficiently over time; nevertheless, they show diminished initial efficacy [27]. The results are noteworthy for stored-product insect management, as water adsorption can reduce the ability of silica to abrade insect cuticles, hence decreasing its desiccation effectiveness [28]. Talc has low-adsorption substances, which may provide more dependable performance in high humidity. Water adsorption isotherms on various silicas have been extensively studied to understand surface properties and porosity [29].

### 3.5. Effects of Sipernat 22 on Sitophilus oryzae and Tribolium castaneum at Two Grain Moisture Contents and 25 °C

The impact of grain moisture content (12% and 15%) on the insecticidal efficacy of Sipernat 22, an amorphous silica dust, against *T. castaneum* and *S. oryzae* at 25 °C was evaluated. The data indicate that at 12% grain moisture, Sipernat 22 demonstrated superior efficacy against both species, with lower LC_50_ values of 12.5 mg/100 g for *T. castaneum* and 46.6 mg/100 g for *S. oryzae*. Nevertheless, as moisture content rose to 15%, the LC_50_ values markedly escalated to 16.8 and 68.7 mg/100 g, respectively. The fact that they work less well at higher moisture levels aligns with what we know about how silica-based dusts work: they physically wear down the insect’s cuticle and adsorb epicuticular lipids, which dries them out and kills them [28]. When the moisture content of the grain is high, it makes this mechanism less effective by filling the silica particles with water, which makes them less abrasive and less able to bind to lipids [30]. *S. oryzae* had consistently higher LC_50_ values at both moisture levels than *T. castaneum*. In general, grain moisture content significantly affects the insecticidal activity of Sipernat 22. So, it is crucial to maintain low moisture levels for perfect control of insects in stored grains, which sustains the superior quality of the grains and enhances the efficacy of desiccant dusts [31] (Table 4).

### 3.6. Joint Action of Insecticides and Silica on T. castaneum and S. oryzae

At a grain moisture content of 15% and a temperature of 25 °C, the performance of various insecticide dust formulations, both alone and in combination with silica, was evaluated against *T. castaneum* and *S. oryzae*. Malathion formulated on talc (4%) recorded LC_50_ values of 52.3 and 84.7 µg a.i/100 g wheat, while pirimiphos methyl on talc (2%) showed LC_50_ values of 20.1 and 32.1 µg a.i/100 g. When loaded onto silica, however, both insecticides exhibited much stronger toxicity, with malathion showing LC_50_ values of 21.5 and 23.3 µg a.i/100 g, and pirimiphos methyl recording 13.4 and 15.5 µg a.i/100 g, against *T. castaneum* and *S. oryzae*, respectively. The combined formulations (insecticide loaded on silica) therefore demonstrated efficacy that clearly exceeded the expected cumulative effect of each component alone. Silica alone displayed slight toxicity, with an LC_50_ of 16.8 and 68.7 mg (16,800 and 68,700 µg)/100 g against *T. castaneum* and *S. oryzae*. The significant reduction in LC_50_ values of loaded insecticides onto silica cannot be attributed to a simple additive action. Instead, this outcome reflects a true synergistic interaction, whereby the abrasive and desiccant properties of silica enhance cuticle disruption, facilitating insecticide penetration and increasing overall toxicity. The abrasive and desiccant properties of silica particles elevate insect susceptibility to insecticides due to cuticle abrasion and penetration of active ingredients [11]. Pirimiphos methyl was more effective against stored insect pests than malathion, which may be attributed to intrinsic toxicity [8]. Moreover, the grain moisture level of 15% seems to maintain the efficacy of silica, allowing it to provide suitable abrasiveness [32]. Conversely, talc, which is mainly inert and hydrophobic, fails to increase the efficacy of insecticides and reveals noticeably lower performance under the same settings [33]. The findings highlight the usefulness of integrating insecticides with amorphous silica, specifically pirimiphos methyl, to enhance insect pest management in stored wheat under moderately humid conditions. The data also correspond with previous research supporting using inert dusts in integrated insect pest management strategies for stored products [34]. Detailed efficacy data for all tested formulations and combinations are provided in Table 5.

### 3.7. Residual Efficacy of Pirimiphos Methyl, Malathion, and Amorphous Silica Against T. castaneum and S. oryzae

The decline in mortality resulting from bioassaying the treated wheat against *S. oryzae* and *T. castaneum* is shown in Table 6 and Table 7 for periods up to 25 weeks from treatment at 25 °C and 12 and 15% wheat moisture content (w.m.c). The residual long effects of malathion, pirimiphos methyl (8.0 mg/kg and 4.0 mg/kg), and silica applied at the calculated LC_50_ to *S. oryzae* (0.47 g/kg) were studied over the storage period. The results showed that wheat treated with malathion resulted in less than 100% mortality of *T. castaneum* after 6 weeks (wk) and zero mortality in the 25th week. Silica, pirimiphos methyl dust (on talc), malathion dust, and pirimiphos methyl formulated on silica resulted in 100% mortality of *T. castaneum* at 12% w.m.c. over the whole experimental period, while at 15% w.m.c. malathion dust had less than 100% mortality after 6 weeks and zero mortality in the 25th week. Silica applied at a concentration corresponding to the LC_50_ for *S. oryzae* (25 °C, 12% w.m.c.) resulted in between 40% and 60% mortality of *T. castaneum* over the 25-week experiment. Malathion dust and pirimiphos methyl formulated on silica caused 100% mortality at the 19th week, while pirimiphos methyl dust started to provide less than 100% mortality after the 17th week. The treatment with malathion dust at 12% w.m.c. started to cause less than 100% mortality of *S. oryzae* after 6 weeks, and activity disappeared after 17 weeks.

The wheat treated with silica continued to cause between 45% and 60% mortality over the 25-week observation period. Wheat treated with pirimiphos methyl dust resulted in less than 100% mortality after 14 weeks, but still resulted in 30% mortality at the 25th week. The treatment with malathion dust and pirimiphos methyl formulated on silica started to cause less than 100% mortality after 12 and 17 wk, and resulted in 72 and 92% mortality at the 25th week, respectively, at 15% w.m.c. The malathion treatment caused less than 100% mortality after the first week, and the treatment with pirimiphos methyl behaved similarly after 6 weeks and resulted in only 10 kills after 25 weeks. The treatment with silica still caused between 10% and 20% mortality of *S. oryzae* throughout the experimental period. The efficacy of malathion and pirimiphos methyl formulated on silica decreased sufficiently, leading to less than 100% mortality after 6 and 10 weeks, respectively, and 26 and 47% mortality at the 25th week. The prolonged efficacy observed when malathion and pirimiphos methyl were formulated on silica, compared with either the insecticides or silica applied alone, suggests that the interaction is not a simple cumulative (additive) effect. If the effects were merely cumulative, the residual toxicity would approximate the sum of the individual mortalities. However, the markedly higher persistence and mortality rates recorded in Table 6 and Table 7 clearly demonstrate a synergistic interaction. This synergy is most likely attributable to the abrasive and desiccant properties of silica, which disrupt the insect cuticle and enhance the penetration and stability of the insecticides, thereby maintaining higher long-term efficacy against *T. castaneum* and *S. oryzae*.

In general, mortality values declined over time, with a sharper decrease at 15% moisture content; however, formulations on silica maintained significantly higher efficacy than individual insecticide dusts, particularly at 12% moisture content, with pirimiphos methyl on silica showing the highest persistence, followed by malathion on silica, while malathion dust exhibited the lowest. The wheat treated with insecticide dust showed that malathion has less residual toxicity than either pirimiphos methyl or silica. One of the main disadvantages of malathion as a stored grain insecticide is its short persistence, particularly on grains of high moisture content, and much effort has gone into finding compounds with greater persistence under these conditions. For example, Abo-Elghar et al. [35] and Foong et al. [36] showed that both chlorpyrifos and chlorpyrifos-methyl were more persistent and effective than malathion, and similar results have been found for pirimiphos methyl, methacrifos, etrimfos, and other newer organophosphorus insecticides. The persistence of malathion, pirimiphos methyl, and malathion–pirimiphos methyl formulations on silica was much lower at 15% w.m.c. compared with 12% w.m.c. against *S. oryzae* and *T. castaneum*. The existence of a critical level of moisture content for the persistence of biologically effective residues of organophosphorus compounds has been indicated before. Agarwal [11] suggested a value of 14% w.m.c for malathion residues on wheat. It can be seen from the results that the insecticide formulated on silica lasted or persisted longer than the individual insecticide dust for both insect species at 12 and 15% w.m.c. The reason for the enhanced toxicity of the insecticides when formulated on silica was not investigated. Still, it might be because silica increased the penetration of insecticides into insects by adsorbing some of the epicuticular lipids and then facilitating the penetration of insecticides. This study has demonstrated a gradual reduction in the effectiveness of malathion, pirimiphos methyl, and amorphous silica against *S. oryzae* and *T. castaneum* after 25 weeks of wheat storage at 25 °C, with grain moisture content (12% versus 15%) being a significant factor. Malathion shows fast degradation, especially at 15% moisture, resulting in a total loss of efficacy by week 17, indicating its instability in humid environments, as corroborated by Alnaji [6]. Pirimiphos methyl exhibited superior performance; however, its efficacy diminished after 14–17 weeks when applied in isolation. The formulation, including amorphous silica, achieved a residual efficacy of 92% mortality after 25 weeks, perhaps attributable to silica’s preventive role against environmental deterioration [37]. Amorphous silica alone resulted in modest yet steady mortality rates (40–60% at 12% w.m.c.) and retained some efficacy (10–20%) even at elevated moisture levels, attributed to its desiccant properties and physical mechanism of lethality. The results emphasize the significance of sophisticated formulation and moisture control in maintaining insecticidal effectiveness and illustrate the possibilities of including inert carriers such as silica for enhanced long-term protection of stored grain. Pirimiphos methyl demonstrates greater long-term efficacy against stored grain pests than malathion. At 7.8 ppm, it surpassed malathion (10.4 ppm) in safeguarding wheat from *Tribolium* species during 12 months, exhibiting slower disintegration [38]. Pirimiphos methyl (5–20 ppm) provides comprehensive protection against several stored-product insects for up to 3 months on wheat and corn, equal to malathion at 10 ppm [39,40]. Synthetic amorphous silica has shown potential as an alternative insecticide, with differing efficiency between hydrophobic and hydrophilic forms against *T. castaneum* and *S. oryzae* [41]. Future research should evaluate the potential sublethal effects of silica treatments, such as morphological impairments, behavioral changes, and their subsequent influence on insect reproduction and population dynamics. Such investigations are essential to understand the long-term effectiveness of pure-silica insecticide formulations beyond acute mortality.

## 4. Conclusions

This study demonstrates that amorphous silica dusts, particularly fumed silicas such as Wacker HDK H20, exhibited high insecticidal effectiveness against *S. oryzae* due to their small particle size and large surface area, although increasing grain moisture reduced their efficacy. Insecticides were applied at recommended protectant rates (≤8 ppm) according to FAO guidelines, and residues remained within approved limits (7–8 ppm). As amorphous silica is classified as a GRAS (FAO/WHO), treated grain can be safely processed following standard aeration and cleaning procedures, ensuring no contamination risk to food or feed. Precipitated silicas such as Sipernat 22 showed moderate effectiveness, also proving promising as a carrier in dust formulations; however, elevated moisture levels (12–15%) reduced their efficacy against *T. castaneum* and *S. oryzae*. Combining amorphous silica with insecticides such as malathion and pirimiphos methyl enhanced and prolonged their activity compared to insecticides alone, particularly under dry storage conditions. Importantly, complete (100%) mortality is not always essential; reducing pest populations below the economic threshold can provide effective protection while also delivering ecological and public health benefits by minimizing chemical inputs.

## Figures and Tables

**Figure 1 insects-16-00981-f001:**
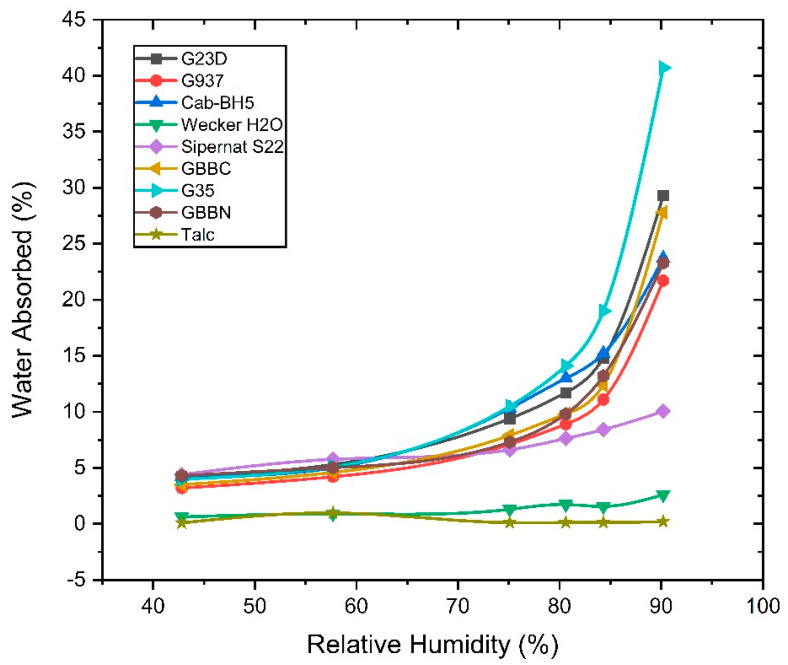
Water adsorption isotherm of various amorphous silicas used for bioassays.

**Table 1 insects-16-00981-t001:** The amorphous silicas used in this study and their physical properties.

Dust Type	Commercial Name	pH 5% Aqueous Suspension	Bulk Density g/100 mL	Specific Surface Area m^2^/g *	Oil Adsorption Capacity g/100 g **	Particle Size Range (µm Primary Particle) ***	Supplier
Fumed silica	Cab-O-Sil EH5	4.2	3.7	390	732	0.007	A
Fumed silica	Wacker HDK H20	4.5	3.5	170	347	0.005	B
Precipitated s.	G35	7.0	15	320	200	2	C
Precipitated s.	GBBN	7.0	15	240	200	8	C
Precipitated s.	G937	7.0	18	320	170	4	C
Precipitated s.	GBBC	7.0	16	240	260	2	C
Precipitated s.	G23D	7.0	31	850	90	5	D
Precipitated s.	Sipernat 22	-	9	190	335	0.2	D
Minerals	Talc	7.4	96	NA	40	~30.4	B

Suppliers: A = Cabot Carbon Limited, Cheshire, UK. B = Great Chemicals Limited, Surrey, UK. C = Joseph Crossfield & Sons Limited, Cheshire, UK. D = Degussa Limited, London, UK. * Manufacturer’s data. ** Risella 17, Shell Chemicals Ltd., London, UK. *** Determined by using Malvern Laser Analyzer, Malvern, UK. NA: Not Applicable.

**Table 2 insects-16-00981-t002:** Effects of different dusts on *S. oryzae* at 12% grain moisture content and 25 °C.

Name of Dust	Regression of N.E.D. Response (y) on Log Dose (x)	LC_50_ (95% Fid. Limits) mg/100 g Wheat	*p* Value
Cab-O-Sil EH5	Y = −5.9 ± 3.7 x	41.0 (36.8, 45.7)	0.11
Wacker HDK H20	Y = −6.1 ± 4.8 x	19.4 (17.2, 21.5)	0.17
G35	Y = −3.3 ± 2.1 x	34.2 (27.5, 43.6)	0.12
GBBN	Y = −4.9 ± 3.1 x	37.1 (29.2, 43.7)	0.09
G937	Y = −6.9 ± 4.4 x	34.6 (32.2, 37.3)	0.09
GBBC	Y = −7.3 ± 4.9 x	31.3 (27.1, 34.9)	0.10
G23D	Y = −6.1 ± 3.8 x	38.3 (35.2, 41.6)	0.51
Sipernat S22	Y = −9.9 ± 5.9 x	46.6 (44.1, 49.2)	0.45
Talc	—	—	—

**Table 3 insects-16-00981-t003:** Effects of different dusts on *S. oryzae* at 15% grain moisture content and 25 °C.

Name of Dust	Regression of N.R.D. Response (y) on Log Dose(x)	LC_50_ (95% Fid. Limits) (mg/100 g Wheat)	*p* Value
Cab-O-Sil EH5	Y = −3.0 + 1.6 x	75.5 (62.1, 91.8)	0.09
Wacker HDK H20	Y = −5.8 + 3.5 x	47.1 (40.5, 54.7)	0.19
G35	Y = −4.0 + 2.4 x	45.8 (38.4, 53.4)	0.64
GBBN	Y = −5.4 + 2.9 x	70.2 (59.8, 82.5)	0.07
G937	Y = −6.6 + 3.5 x	73.1 (66.7, 80.1)	0.77
GBBC	Y = −7.7 + 4.0 x	82.5 (75.7, 89.8)	0.76
G23D	Y = −3.7 + 2.2 x	46.7 (21.0, 75.2)	0.01
Sipernat S22	Y = −8.0 + 4.3 x	73.1 (68.1, 79.7)	0.94
Talc	--	--	--

**Table 4 insects-16-00981-t004:** Effects of Sipernat 22 on *S. oryzae* and *T. castaneum* at two grain moisture contents and 25 °C.

Grain Moisture Content (%)	Name of Insect	Regression of N.B.D. Response (y) on Log Dose (x)	LC_50_ (95% Fid. Limits) mg/100 g Wheat	*p* Value
12	*T*. *castaneum*	Y = −7.5 ± 6.8 x	12.5 (11.5, 13.7)	0.00
	*S. oryzae*	Y = −9.9 ± 4.9 x	46.6 (44.1, 49.2)	0.45
15	*T. castaneum*	Y = −8.7 ± 7.1 x	16.8 (16.1, 17.6)	0.54
	*S. oryzae*	Y = −7.8 ± 4.2 x	68.7 (63.1, 74.7)	0.11

**Table 5 insects-16-00981-t005:** Effects of grain moisture content (15%) on joint action of insecticides and silica on *T. castaneum* and *S. oryzae* at 25 °C.

Name of Insect	Formulation	Regression of N.B.D. Response (y) on Log Dose (x)	LC_50_ (95% Fid. Limits) µg a.i./100 g Wheat	*p* Value
*T. castaneum*	Malathion on talc (4%)	Y = −0.8 ± 7.4 x	52.3 (45.2, 59.4)	0.29
	Malathion on silica (0.2%)	Y = −10.2 ± 9.8 x	21.5 (20.3, 22.6)	0.45
	Pirimiphos methyl on talc (2%)	Y = −0.0 ± 13.8 x	20.1 (18.9, 21.9)	0.22
	Pirimiphos methyl on silica (0.1%)	Y = −6.3 ± 5.7 x	13.4 (12.2, 14.3)	0.45
*S. oryzae*	Malathion on talc (4%)	Y = −2.9 ± 9.3 x	84.7 (80.5, 88.2)	0.23
	Malathion on silica (0.2%)	Y = −3.6 ± 3.4x	23.3(21.4, 25.5)	0.20
	Pirimiphos methyl on talc (2%)	Y = −1.3 ± 5.7 x	32.1 (30.2, 34.7)	0.35
	Pirimiphos methyl on silica (0.1%)	Y = −4.1 ± 3.4 x	15.5 (14.3, 16.6)	0.22

**Table 6 insects-16-00981-t006:** Toxicity of malathion, pirimiphos methyl, and silica formulations against *T. castaneum* after incubation at 25 °C for different time intervals.

Grain Moisture Content	Formulation	Mean Percent of Kill at Time Intervals from Application (Week)								
		0 WK	6 WK	10 WK	12 WK	14 WK	17 WK	19 WK	22 WK	25 WK
12%	Malathion (8 ppm) on talc	100 ^a^ ± 0.0	100 ^a^ ± 0.0	94.7 ^b^ ± 2.1	66 ^b^ ± 1.21	16.3 ^b^ ± 1.09	11.3 ^b^ ± 1.13	10.7 ^b^ ± 0.51	7.3 ^b^ ± 0.45	0.0 ^b^ ± 0.0
	Pirimiphos methyl (4 ppm) on talc	100 ^a^ ± 0.0	100 ^a^ ± 0.0	100 ^a^ ± 0.0	100 ^a^ ± 0.0	100 ^a^ ± 0.0	100 ^a^ ± 0.0	100 ^a^ ± 0.0	100 ^a^ ± 0.0	100 ^a^ ± 0.0
	Malathion on silica	100 ^a^ ± 0.0	100 ^a^ ± 0.0	100 ^a^ ± 0.0	100 ^a^ ± 0.0	100 ^a^ ± 0.0	100 ^a^ ± 0.0	100 ^a^ ± 0.0	100 ^a^ ± 0.0	100 ^a^ ± 0.0
	Pirimiphos methyl on silica	100 ^a^ ± 0.0	100 ^a^ ± 0.0	100 ^a^ ± 0.0	100 ^a^ ± 0.0	100 ^a^ ± 0.0	100 ^a^ ± 0.0	100 ^a^ ± 0.0	100 ^a^ ± 0.0	100 ^a^ ± 0.0
	Silica (0.47 g/kg)	100 ^a^ ± 0.0	100 ^a^ ± 0.0	100 ^a^ ± 0.0	100 ^a^ ± 0.0	100 ^a^ ± 0.0	100 ^a^ ± 0.0	100 ^a^ ± 0.0	100 ^a^ ± 0.0	100 ^a^ ± 0.0
	LSD 5%	-	-	1.777	0.985	0.918	0.951	0.429	0.379	-
15%	Malathion (8 ppm)	100 ^a^ ± 0.0	100 ^a^ ± 0.0	9.3 ^c^ ± 0.44	10.7 ^c^ ± 0.57	13.3 ^d^ ± 0.22	1.3 ^d^ ± 0.06	0.7 ^d^ ± 0.05	0.0 ^d^ ± 0.0	0.0 ^d^ ± 0.0
	Pirimiphos methyl (4 ppm)	100 ^a^ ± 0.0	100 ^a^ ± 0.0	100 ^a^ ± 0.0	100 ^a^ ± 0.0	100 ^a^ ± 0.0	100 ^a^ ± 0.0	94.7 ^b^ ± 2.41	92.7 ^b^ ± 4.92	92 ^b^ ± 5.21
	Malathion on silica	100 ^a^ ± 0.0	100 ^a^ ± 0.0	95.3 ^a^ ± 3.71	100 ^a^ ± 0.0	94 ^b^ ± 3.31	94 ^b^ ± 3.83	94 ^b^ ± 4.66	96.7 ^ab^ ± 1.49	96.7 ^ab^ ± 1.68
	Pirimiphos methyl on silica	100 ^a^ ± 0.0	100 ^a^ ± 0.0	100 ^a^ ± 0.0	100 ^a^ ± 0.0	100 ^a^ ± 0.0	100 ^a^ ± 0.0	100 ^a^ ± 0.0	100 ^a^ ± 0.0	100 ^a^ ± 0.0
	Silica (0.47 g/kg)	39.3 ^b^ ± 2.70	36.6 ^b^ ± 2.10	60.7 ^b^ ± 5.40	53.3 ^b^ ± 3.70	46 ^c^ ± 2.90	38 ^c^ ± 2.29	50 ^c^ ± 3.17	56.7 ^c^ ± 4.36	58.6 ^c^ ± 4.29
	LSD 5%	2.273	1.768	5.529	3.152	3.710	3.758	5.162	5.676	5.856

Values are means of three replicates ± SD. At each grain moisture level, different letters within the same column indicate statistically significant differences among treatments at the 5% level (LSD test), while identical letters denote no significant differences.

**Table 7 insects-16-00981-t007:** Toxicity of malathion, pirimiphos methyl, and silica formulations against *S. oryzae* after incubation at 25 °C for different time intervals.

Grain Moisture Content	Formulation	Mean Percent of Kill at Time Intervals from Application (Week)								
		0 WK	6 WK	10 WK	12 WK	14 WK	17 WK	19 WK	22 WK	25 WK
12%	Malathion (8 ppm)	100 ^a^ ± 0.0	100 ^a^ ± 0.0	74.7 ^b^ ± 3.33	32.7 ^b^ ± 1.94	16.7 ^c^ ± 1.58	0.0 ^e^ ± 0.0	0.0 ^d^ ± 0.0	0.0 ^e^ ± 0.0	0.0 ^e^ ± 0.0
	Pirimiphos methyl (4 ppm)	100 ^a^ ± 0.0	100 ^a^ ± 0.0	100 ^a^ ± 0.0	99.3 ^a^ ± 5.67	100 ^a^ ± 0.0	65.3 ^c^ ± 3.26	63.3 ^b^ ± 5.83	33.3 ^d^ ± 2.75	30.7 ^d^ ± 1.57
	Malathion on silica	100 ^a^ ± 0.0	100 ^a^ ± 0.0	100 ^a^ ± 0.0	96 ^a^ ± 4.49	98 ^a^ ± 4.21	84.7 ^b^ ± 3.48	57.3 ^bc^ ± 2.49	70.7 ^b^ ± 3.25	72.7 ^b^ ± 3.19
	Pirimiphos methyl on silica	100 ^a^ ± 0.0	100 ^a^ ± 0.0	100 ^a^ ± 0.0	98.7 ^a^ ± 5.92	100 ^a^ ± 0.0	100 ^a^ ± 0.0	94 ^a^ ± 4.37	93.3 ^a^ ± 6.18	92.7 ^a^ ± 5.99
	Silica (0.47 g/kg)	57 ^b^ ± 2.48	58 ^b^ ± 1.94	60.7 ^c^ ± 3.42	53.3 ^c^ ± 3.71	57.3 ^b^ ± 3.48	59.3 ^d^ ± 3.13	56 ^c^ ± 2.66	59.3 ^c^ ± 4.68	60.7 ^c^ ± 4.19
	LSD 5%	2.088	1.634	4.019	8.623	4.787	4.803	6.859	7.447	6.843
15%	Malathion (8 ppm)	100 ^a^ ± 0.0	64.7 ^b^ ± 3.53	31.3 ^b^ ± 2.47	4 ^d^ ± 0.30	0.0 ^d^ ± 0.0	0.0 ^e^ ± 0.0	0.0 ^d^ ± 0.0	0.0 ^e^ ± 0.0	0.0 ± 0.0
	Pirimiphos methyl (4 ppm)	100 ^a^ ± 0.0	100 ^a^ ± 0.0	98.7 ^a^ ± 4.92	96.7 ^a^ ± 3.60	99.3 ^a^ ± 4.40	27.3 ^c^ ± 1.70	25.3 ^b^ ± 0.60	4 ^d^ ± 0.79	10.6 ^d^ ± 1.85
	Malathion on silica	100 ^a^ ± 0.0	100 ^a^ ± 0.0	97.3 ^a^ ± 2.99	99.3 ^a^ ± 4.93	82 ^b^ ± 3.94	46.7 ^b^ ± 2.16	42 ^a^ ± 3.54	21.3 ^b^ ± 1.27	26.7 ^b^ ± 2.02
	Pirimiphos methyl on silica	100 ^a^ ± 0.0	100 ^a^ ± 0.0	100 ^a^ ± 0.0	83.3 ^b^ ± 2.50	99.3 ^a^ ± 4.62	55.3 ^a^ ± 3.13	45.7 ^a^ ± 3.93	40.7 ^a^ ± 2.98	47.3 ^a^ ± 3.48
	Silica (0.47 g/kg)	16.7 ^b^ ± 2.09	15.3 ^c^ ± 1.11	18 ^c^ ± 2.18	15.3 ^c^ ± 3.09	18.7 ^c^ ± 1.78	16.7 ^d^ ± 1.39	15.3 ^c^ ± 2.44	16 ^c^ ± 1.73	18.7 ^c^ ± 2.18
	LSD 5%	1.757	3.116	5.588	6.134	6.489	3.693	4.936	3.167	4.154

Values are means of three replicates ± SD. At each grain moisture level, different letters within the same column indicate statistically significant differences among treatments at the 5% level (LSD test), while identical letters denote no significant differences.

## Data Availability

The original contributions presented in this study are included in the article. Further inquiries can be directed to the corresponding author.

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
