# Peer review of "Relative Effectiveness of Amorphous Silica, Malathion, and Pirimiphos Methyl in Controlling Sitophilus oryzae and Tribolium castaneum and Their Long-Term Effects on Stored Wheat Under Laboratory Conditions"

_insects, 2025, doi:10.3390/insects16090981_

Round 1
Reviewer 1 Report (Previous Reviewer 1)
Comments and Suggestions for Authors
Dear editor and authors,
The article has been revised according to the suggested revisions. All of my revision suggestions have been implemented. There are still some spelling mistakes. The article is acceptable after minor revisions.
Best regards
Author Response
Comment 1: The article has been revised according to the suggested revisions. All of my revision suggestions have been implemented. There are still some spelling mistakes.
Response 1: Thank you so much for the valuable note. We have modified the revised manuscript after using a professional language editing service, and a certificate is attached as a supplement file.

Reviewer 2 Report (New Reviewer)
Comments and Suggestions for Authors
I have read and revised the manuscript “Relative Effectiveness of Amorphous Silica, Malathion, and Pirimiphos-methyl in Controlling Sitophilus oryzae and Tribolium castaneum and Their Long-Term Effects on Stored Wheat”.
Overall, the experimental activities carried out are quite interesting, even if not totally innovative because various indications in this regard can be found in other international publications. However, the annotations and suggestions that I have inserted directly in the attached PDF may improve and enhance the work, but it is important that the Authors make the necessary changes requested. In my opinion it is necessary to add in the title that the research was carried out in the laboratory under controlled conditionsIt is necessary to specify whether the results obtained are a synergistic effect between different substances or whether it is a cumulative effect between different substances.Another important practical aspect is the destination of the treated grains, whether they will be stored for a long time or processed by the food industry or consumed by humans or by animals in a short time. These specifications are the basis of the choices that technicians will have to make in operational practice, before suggesting the use of any biocide

Author Response
Comment 1: Add in laboratory conditions (page 1)
Response 1: We appreciate the reviewer’s constructive suggestion. The sentence has been revised and added to the Title and line (25)
Comment 2: Highlighted Text (Page 2), Line 53
Response 2: Thanks, it has been done
Comment 3: Highlighted Text (Page 2), Line 54
Response 3: Thanks, it has been done
Comment 4: These two paragraphs can be shortened at line 58
Response 4: Thank you for the comment. We have shortened the two paragraphs to remove redundancies while keeping the key points on wheat’s importance and the destructive roles of S. oryzae and T. castaneum (Lines: 43-53).
Comment 5: Highlighted Text (Page 2), line 79
Response 5: Thanks, it has been done
Comment 6: Highlighted Text (Page 2), Line 80
Response 6: Thanks, it has been done
Comment 7: It is better not to cite commercial names in scientific publications.
Response 7: We agree with you. We have removed the company/manufacturer names of silica (Lines: 102-105).
We would like to clarify that malathion and pirimiphos-methyl are not commercial or trade names but internationally recognized common names of the active ingredients. Trade names (e.g., Cythion® for malathion and Actellic® for pirimiphos-methyl) vary by manufacturer, while our manuscript consistently uses the common names for scientific accuracy and clarity. (Lines: 107-108).
Comment 8:
Highlighted Text 127, 138-139
Response 8:
Thanks, it has been done
Comment 9:
In case of silica…effectiveness must take into account not only death but also the morphological impairments practiced that will influence subsequent generations.
Response 9:
We appreciate the reviewer’s insightful comment. We fully agree that the effectiveness of silica-based dusts may extend beyond immediate mortality to include sublethal effects such as morphological impairments, which could influence survival, reproduction, and subsequent generations. However, the current study was primarily designed to evaluate adult mortality as the main indicator of efficacy. We acknowledge this important point and will highlight in the discussion that future research should investigate potential sublethal effects, including morphological impairments and their impact on population dynamics.
Comment 10:
In this part it is very important to be able to differentiate the synergistic effect from the cumulative effect of the two biocide systems adopted (insecticide and silica)
3.6. Joint action of insecticides and silica on T. castaneum and S. oryzae) line (283)
Response 10:
Tables 4 and 5. present the results of each insecticide and silica tested separately, as well as the mixtures (insecticide formulated on silica). The data clearly show that the formulations containing insecticides loaded on silica exhibited significantly higher efficacy compared with each component alone. These marked differences indicate a synergistic effect, rather than a simple cumulative action.
Comment 11:
In this part it is very important to be able to differentiate the synergistic effect from the cumulative effect of the two biocide systems adopted (insecticide and silica) line (337)
Response 11:
Tables 6 and 7 present the results of insecticides and silica tested individually, as well as their combinations (insecticides loaded on silica). The data demonstrate that the formulations with insecticides carried on silica achieved significantly higher efficacy compared to each component alone, indicating a synergistic effect rather than a simple cumulative action. Each mortality measurement represents an independent bioassay; wheat was treated in 4 kg batches for each treatment, and separate 100 g samples were stored at 25 °C. At each time interval, new samples (three replicates × 100 g) were taken, with 50 adult insects introduced per replicate, and mortality was recorded after 72 hours. Therefore, the observed decline in mortality over time reflects the reduction in residual efficacy of the treatment, not the survival or accumulation of previously exposed insects. line (158-163).
Comment 12:
An important consideration is the fate of the treated grain: how long will it take to process it? This could potentially contaminate the product intended for human or animal consumption.
It is not important to reach 100% effectiveness, sometimes it is enough to get below the economic threshold with important ecological and general health gains for the product and consumers.
Response 12:
Thank you for this valuable observation. Indeed, the insecticides used in this study are among the safest and most widely approved compounds for stored-grain pest management. The concentrations applied were relatively low, and their combination with silica resulted in a marked improvement in efficacy while reducing the need for higher insecticide concentrations. It is also important to note that these insecticides have relatively short persistence, which minimizes the risk of residue accumulation in food products. This makes them suitable as safe and approved candidates for use in stored-product pest control programs while remaining within internationally accepted limits and food safety standards.
Comment 13:
It is not important to reach 100% effectiveness; sometimes it is enough to get below the economic threshold with important ecological and general health gains for the product and consumers.
Response 13:
Thank you for this valuable comment. We agree that in many pest management programs it is not always necessary to achieve 100% mortality, and that reducing pest populations below the economic threshold can indeed provide significant ecological and public health benefits. However, this principle does not apply equally to all stored-product pests. Some species, such as Sitophilus oryzae and Trogoderma granarium, are destructive even at low infestation levels, causing economic losses, thereby necessitating more stringent control measures in such cases.

Round 2
Reviewer 2 Report (New Reviewer)
Comments and Suggestions for Authors
The Authors did not accept all the suggestions and comments reported in my previous
revision. I encourage them to be more forthcoming and make the requested changes.
Otherwise, I cannot express a fully positive opinion.
Author Response
Comment 1: Add in laboratory conditions (page 1)
Response 1: We appreciate the reviewer’s constructive suggestion. The sentence has been revised and added to the Title and line (25)
Comment 2: Highlighted Text (Page 2), Line 53
Response 2: Thanks, it has been done
Comment 3: Highlighted Text (Page 2) line 54
Response 3: Thanks, it has been done
Comment 4: These two paragraphs can be shortened at line 58
Response 4: Thank you for the comment. We have shortened the two paragraphs to remove redundancies while keeping the key points on wheat’s importance and the destructive roles of S. oryzae and T. castaneum (Lines: 43-53).
Comment 5: Highlighted Text (Page 2), Line 79
Response 5: Thanks, it has been done
Comment 6: Highlighted Text (Page 2), Line 80
Response 6: Thanks, it has been done
Comment 7: It is better not to cite commercial names in scientific publications.
Response 7: We agree with you. We have removed the company/manufacturer names of silica (Lines: 102-105).
We would like to clarify that malathion and pirimiphos-methyl are not commercial or trade names but internationally recognized common names of the active ingredients. Trade names (e.g., Cythion® for malathion and Actellic® for pirimiphos-methyl) vary by manufacturer, while our manuscript consistently uses the common names for scientific accuracy and clarity. (Lines: 107-108).
Comment 8:
Highlighted Text 127, 138-139
Response 8:
Thanks, it has been done
Comment 9:
In the case of silica, effectiveness must take into account not only death but also the morphological impairments caused, which will influence subsequent generations.
Response 9:
We appreciate the reviewer’s insightful comment. We fully agree that the effectiveness of silica-based dusts may extend beyond immediate mortality to include sublethal effects such as morphological impairments, which could influence survival, reproduction, and subsequent generations. However, the current study was primarily designed to evaluate adult mortality as the main indicator of efficacy. We acknowledge this important point and will highlight in the discussion that future research should investigate potential sublethal effects, including morphological impairments and their impact on population dynamics.
Comment 10:
In this part, it is very important to be able to differentiate the synergistic effect from the cumulative effect of the two biocide systems adopted (insecticide and silica)
3.6. Joint action of insecticides and silica on T. castaneum and S. oryzae
Response 10:
We sincerely thank the reviewer for this valuable comment. In the revised manuscript (Section 3.6), we have clarified the distinction between the cumulative and synergistic effects. The LCâ‚…â‚€ values presented in Tables 4 and 5 show that insecticides formulated on silica exhibited much higher efficacy than when applied alone. For instance, malathion on talc recorded LCâ‚…â‚€ values of 52.3 and 84.7 µg a.i/100 g wheat against T. castaneum and S. oryzae, respectively, whereas malathion on silica reduced the LCâ‚…â‚€ values to 21.5 and 23.3 µg a.i/100 g. Similarly, pirimiphos-methyl on silica showed significantly greater toxicity compared to pirimiphos-methyl on talc. Since silica alone showed slight toxicity with LCâ‚…â‚€ values of 16.8 and 68.7mg (16800 and 68700 µg)/100g, the increase in LCâ‚…â‚€ values reflects the very low toxicity, while the significant reduction in LCâ‚…â‚€ values of loaded insecticides onto silica cannot be explained by a simple additive effect. Instead, they indicate a true synergistic interaction, likely due to the abrasive and desiccant properties of silica. line (277-289)
Comment 11:
In this part, it is very important to be able to differentiate the synergistic effect from the cumulative effect of the two biocide systems adopted (insecticide and silica)
Response 11:
We sincerely thank the reviewer for this important comment. In the revised manuscript (Section 3.7), we have clarified the difference between cumulative and synergistic effects. As shown in Tables 6 and 7, formulations of malathion and pirimiphos-methyl on silica maintained much higher residual efficacy against T. castaneum and S. oryzae compared to insecticides or silica alone. Since silica alone caused only modest and declining mortality, the enhanced persistence observed cannot be attributed to a simple cumulative effect. Instead, the much higher and prolonged efficacy of the combined formulations indicates a true synergistic interaction, likely due to the abrasive and desiccant properties of silica that facilitate insecticide penetration. Tables (4 and 5) support the results. line (332-341)
Comment 12:
An important consideration is the fate of the treated grain: how long will it take to process it? This could potentially contaminate the product intended for human or animal consumption.
Response 12:
We sincerely thank the reviewer for this insightful comment. In response, we have clarified in the revised manuscript that the insecticides in our study were applied at recommended protectant rates (≤8 ppm), consistent with FAO guidelines (10 ppm for malathion and 8 ppm for pirimiphos-methyl). Silica, used as a carrier, is generally recognized as safe (GRAS) by FAO/WHO. Importantly, treated grain can be adequately aerated and cleaned before processing, ensuring residues remain below maximum residue limits (7–8 ppm). Lines 401-408
Comment 13:
It is not important to reach 100% effectiveness; sometimes it is enough to get below the economic threshold with important ecological and general health gains for the product and consumers.
Response 13:
We sincerely appreciate the reviewer’s insightful comment. In the revised manuscript, we clarified that the objective of stored-product protection is not always to achieve 100% mortality. Instead, reducing pest populations below the economic threshold is often sufficient to prevent significant damage. However, species such as Sitophilus oryzae and Trogoderma granarium are destructive even at low infestation levels, causing economic losses, and necessitating more stringent control measures in such cases. Accordingly, we have amended the conclusions to reflect this important perspective, lines 408-415.

Round 3
Reviewer 2 Report (New Reviewer)
Comments and Suggestions for Authors The Authors of the manuscript have answered the reviewers' questions,provided clarifications, and revised some sections of the original text.
The new version of the work is quite interesting and can be considered for publication.
This manuscript is a resubmission of an earlier submission. The following is a list of the peer review reports and author responses from that submission.
Round 1
Reviewer 1 Report
Comments and Suggestions for Authors
The authors have compared the effectiveness of silica, malathion, and pirimiphos-methyl on Sitophilus oryzae and Tribolium castaneum. I have reviewed this ms and it needs improvements before it is accepted for publication. The authors did not mention the number of treatments and their replications, which affect the reproducibility of data. Further comments and suggestions for improvement of the manuscripts are given below.
L21-22: Revise this phrase “This can help farmers and storage facilities reduce food 21 waste and support safer, more environmentally friendly pest management practices”
L24: “Sitophilus oryzae and Tribolium castaneum”.. Must be in Italic with authority name when written first time in text (family, order
L42: “Triticum aestivum”.. must be italic. Also all species names have to italic in all MS.
L77: numbered as “2.1.1. Insect cultures”
L78: Just write “S. oryzae and T. castaneum”, “Tribolium. Castaneum” has wrong writing
L80: Its better not to start a line with abbreviated name.,”S. oryzae”
L83: “..Tranferred” How much quantity?, which dimensiions culture jars
L99: “unsexed adults” How did you kept these insects for 3 weeks without mating? Male and female differentiation? I think this words hqve to Change as “mixed gender”
L100: Heading numbers revised accordingly. Also remove colon in headings
L101-102: Write chemical compositions and company name after each product respectively.
L: “Long term effect of pirimiphos methyl, malathion, and amorphous silica on T.casta- 148 neum and S.oryzae.
L133: “Unsexed” what do you mean by unsexed? either mixed sexes or unmated adults?
L148 :” Long term effect of pirimiphos methyl, malathion, and amorphous silica on T.castaneum and S.oryzae.” no meaning of this
L171: Its better to start with your results and add table caption at end of paragraph. i.e. (Table X)
L191” Its better to start with your results and add table caption at end of paragraph. i.e. (Table X)
L193: “The data” change as “After three days exposure, the data..”
L214-218: Delete it. repeat. “The efficacy of fumed sili……………. regression fit.
L221-“very particles” means?
L 222- “which enhance insect mortality 222 through cuticle abrasion and dehydration” add reference
“
L225: “Zing Zing, et al. 18” 18 showing uppercase , not in the bracket
L235: “on Table 3” What is P ? How did you calculate it? What is difference between 0.09 for LC50 value of 75.5 and P = 0.94 for 73.1.?
L276: “…Sitophilus. Oryzae and Tribolium. Castaneum…” need correction of writing
L 279: Its better to start with your results and add table caption at end of paragraph. i.e. (Table X)
L 300: “T. Castaneum and S. Oryzae at 25 °C.” need correction of writing also in table
Reviewer 2 Report
Comments and Suggestions for Authors
Normally, the pesticides should not be mixed with grain or should not contact with grain. How can you add pesticides with silica and mix with grain. How will you separate the pesticides which are adhered from grain? it may mixed with seed.
During culturing of insects, the insects must be removed after 24hrs of egg laying to get uniform aged adults. If you allow for 2 weeks, it is not possible to get uniform aged adults.
If you get 100 per cent mortality in first few weeks, how can you get lesser percentage of mortality after few weeks.
The experiment was conducted at 15% moisture content. What is the normal moisture content of wheat during storage?
The statistical analysis for Table 6 and 7, is not given.